# Learning to Route LLMs from Bandit Feedback: One Policy, Many Trade-offs

## Abstract

Efficient use of large language models (LLMs) is critical for deployment at scale: without adaptive routing, systems either overpay for strong models or risk poor performance from weaker ones. Selecting the right LLM for each query is fundamentally an online decision problem: models differ in strengths, prices fluctuate, and users value accuracy and cost differently. Yet most routers are trained offline with labels for all candidate models, an assumption that breaks in deployment, where only the outcome of the chosen model is observed. We bridge this gap with BaRP, a Bandit-feedback Routing with Preferences approach that trains under the same partial-feedback restriction as deployment, while supporting preference-tunable inference: operators can dial the performance–cost trade-off at test time without retraining. Framed as a contextual bandit over prompt features and a user preference vector, our method simulates an online feedback setting during training and adapts its routing decisions to each new prompt, rather than depending on full-information offline supervision. Comprehensive experiments show that our method consistently outperforms strong offline routers by at least 12.46% and the largest LLM by at least 2.45%, and generalizes robustly for unseen tasks.

## 1 Introduction

Large language models (LLMs) vary substantially in their strengths, weaknesses, and operating costs. No single model dominates across all prompts and tasks, and both pricing and quality change over time. Users and applications also vary in how they prioritize accuracy and cost. At deployment scale, a system must therefore decide *per query* which model to call under a performance–cost trade-off. A common solution is to employ a *router*, a learned policy that selects an LLM for each incoming prompt. The challenge is that, once deployed, the router only receives feedback from the model it actually calls: it observes the accuracy and cost of the selected model but learns nothing about the alternatives. This setting,

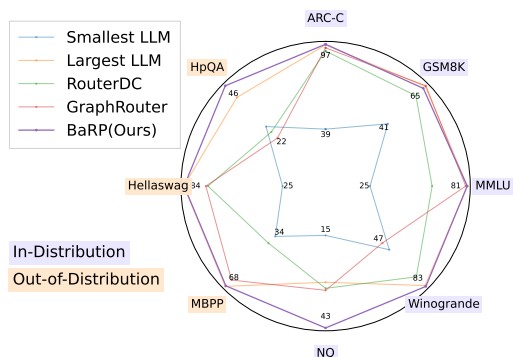

Figure 1: Testing score of baselines and BaRP on in-distribution and out-of-distribution tasks.

where supervision is restricted to the chosen action, is known as *bandit feedback*. In contrast, most existing routers are trained offline with labels for all candidate models on every prompt, creating a mismatch between training and deployment.

Prior work illustrates two recurring limitations. The first is the reliance on *full-information offline supervision*, where training requires labels from all candidate LLMs on each prompt. For example, RouterDC (Chen et al., 2024) compares every prompt across multiple LLM outputs, so it cannot be trained once deployed, when only the chosen model's feedback is available. GraphRouter (Feng et al., 2025) faces the same limitation, as it learns graph-structured representations that rely on full-information labels. The second limitation is the lack of *preference-tunable inference*, the ability to adjust routing at test time to reflect user-specified performance–cost trade-offs without retraining. For instance, RouterDC (Chen et al., 2024) yields a routing policy tied to the trade-off during

Table 1: Comparison of routing methods. "Full-information Offline Supervision" indicates that training requires labels from all candidate LLMs for each prompt. "Preference-tunable Inference" refers to whether the method can adjust routing at test time to accommodate user-specified performance–cost trade-offs without retraining.

| Method | Full-information Offline Supervision | Preference-tunable Inference |
|---|---|---|
| GraphRouter (Feng et al., 2025) | Required | No |
| RouterDC (Chen et al., 2024) | Required | No |
| C2MAB-V (Dai et al., 2024) | Not required | No |
| MAR (Zhang et al., 2025) | Not required | No |
| LLM Bandit (Li, 2025) | Required | Yes |
| **BARP (Ours)** | **Not required** | **Yes** |

training, GraphRouter (Feng et al., 2025) supports only three predefined scenarios and is therefore not fully preference-tunable, while our method can shift its choices depending on whether a user prioritizes performance or cost. Bandit-style approaches such as C2MAB-V (Dai et al., 2024) and Multi-Armed Router (MAR) (Zhang et al., 2025) avoid full-information supervision but still lack this controllability, and LLM Bandit (Li, 2025) introduces preferences but relies on offline pre-training that assumes full labels. Table 1 summarizes these methods across the two dimensions of supervision and controllability. Additional related work is discussed in Section 5.

We propose BARP, a Bandit-feedback Routing with Preferences framework that addresses both limitations in a unified manner. Our formulation treats routing as a *multi-objective contextual bandit* problem: the router must balance two competing objectives, performance and cost, given only bandit feedback. To capture user preferences, we condition the policy on a trade-off vector that specifies the relative importance of performance and cost. The router encodes each prompt together with this vector and outputs a distribution over candidate LLMs. The policy is trained with policy-gradient updates regularized by entropy for exploration and stabilized by calibrated cost scaling. This design removes the need for labels from all models during training while allowing operators to adjust performance–cost preference at inference without retraining. By aligning training with the partial-feedback setting of deployment and providing controllability at test time, BARP offers a practical solution for real-world routing.

In summary, our main contributions are as follows:

- We formulate multi-objective LLM routing as a contextual bandit problem in which the router learns from bandit feedback while conditioning on a user preference vector that specifies the trade-off between accuracy and cost. This formulation eliminates the need for full supervision across all candidate models and enables per-request controllability.

- We design a routing policy that integrates prompt representations with the preference vector, and train it using entropy-regularized policy gradients with calibrated cost scaling, which encourages exploration and ensures stable optimization under partial feedback.

- We validate our framework on RouterBench and two question-answering datasets, demonstrating significant performance gains over strong baselines. On **in-distribution** tasks, our method surpasses the top-performing individual LLM by 3.81% and full-information offline routers by 12.46%. On **out-of-distribution** tasks, the gains are 2.45% and 25.99% respectively, as shown in Fig. 1.

## 2 APPROACH

We present BARP, a **Ba**ndit–feedback **R**outer with **P**references. The core idea is to treat routing as a *multi-objective contextual bandit*: the router balances performance and cost while observing feedback only for the selected model. This section introduces the problem setting (Sec. 2.1), then defines the policy architecture (Sec. 2.2), followed by the objective and learning procedure (Sec. 2.3). The training and inference procedures are provided in Algorithm 1 and Sec. 2.4. For intuition, Fig. 2 illustrates a single request in the training process: a prompt and a user preference enter the router, which selects an LLM, receives bandit feedback, and updates the policy.

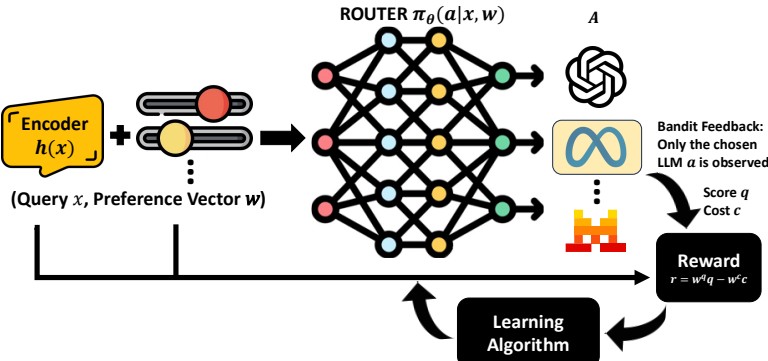

Figure 2: The training pipeline of BARP. The router takes the context (query $x_t$ and preference $w_t$) and selects an LLM. It then receives bandit feedback (the score and cost of the chosen LLM only) to calculate a reward $r_t$. This reward drives a **learning algorithm** to update the router's parameters, including policy gradient methods like REINFORCE (Sec. 2.3) and classic bandit algorithms such as LinUCB, Thompson Sampling, and $\epsilon$-greedy (Sec. 4.6).

## 2.1 PROBLEM SETTING

We formally define the preference-conditioned LLM routing task as a contextual bandit problem. In each round $t$, an agent observes a context and selects an arm, receiving a reward based on its choice. The **Context** ($s_t$) is a tuple $s_t = (x_t, w_t)$, where $x_t$ is the input prompt and $w_t = (w_t^q, w_t^c)$ is a user preference vector on the 1-simplex. Here, $w_t^q$ represents the weight the user places on the performance score, while $w_t^c$ represents the weight on minimizing cost. The set of $K$ available LLMs constitutes the **Arms** ($A$), or the action space $\{1, \ldots, K\}$. The router selects an **Action** ($a_t$) from this set, corresponding to choosing a single LLM to process the prompt. Upon selection, the router receives a scalar **Reward** ($r_t$) based on bandit feedback for the chosen arm. This reward combines the two objectives according to the user's preference:

$$r_t = w_t^q \, q_t - w_t^c \, \tilde{c}_t, \quad \text{where} \quad \tilde{c}_t = \min\left(\frac{c_t}{\tau}, 1\right). \quad (1)$$

where the score $q_t$ is a task-appropriate metric scaled to $[0, 1]$, $\tau > 0$ caps cost $c_t$ so that score and (normalized) cost are on comparable scales. The overall goal is to learn a policy that maximizes the expected cumulative reward.

## 2.2 POLICY ARCHITECTURE

The routing policy $\pi_\theta(a \mid s)$ is a neural network that maps a context $s = (x, w)$ to a probability distribution over the $K$ LLMs. The architecture is composed of three sequential components. First, a **Prompt Encoder**, a frozen pre-trained sentence transformer $h$, encodes the prompt $x$ into a semantic vector $h(x) \in \mathbb{R}^{d_e}$. Second, a **Preference Encoder**, a small multilayer perceptron (MLP) $\phi$, maps the 2-dimensional preference vector $w$ into a higher-dimensional embedding $\phi(w) \in \mathbb{R}^{d_p}$. Finally, the prompt and preference embeddings are concatenated to form a joint representation, $z = [h(x); \phi(w)]$, which is passed to a **Decision Head**, $g_\theta$, to produce logits $o \in \mathbb{R}^K$. The final policy is obtained by applying a softmax function to these logits:

$$\pi_\theta(a \mid x, w) = \text{softmax}(g_\theta(z))_a = \frac{\exp(o_a)}{\sum_{a'=1}^{K} \exp(o_{a'})}. \quad (2)$$

During training we sample $a_t \sim \pi_\theta(\cdot \mid x_t, w_t)$ to ensure exploration. At inference, we output the deterministic choice:

$$a^*(x, w) = \arg\max_{a \in A} \pi_\theta(a \mid x, w), \quad (3)$$

---

**Algorithm 1** The Training and Inference Procedure for BARP.

---

1: **Inputs:** encoder $h$, preference MLP $\phi$, head $g_\theta$; cost cap $\tau$; entropy coeff $\beta$.
2: Initialize parameters $\theta$.
3: **for** $t = 1$ to $T$ **do**
4:     Receive prompt $x_t$ and sample preference $w_t$ (random on the 1-simplex).
5:     Compute $h_t \leftarrow h(x_t)$ and $u_t \leftarrow \phi(w_t)$; form $z_t \leftarrow [h_t; u_t]$.
6:     $o_t \leftarrow g_\theta(z_t), \quad \pi_t \leftarrow \text{softmax}(o_t)$.
7:     Sample $a_t \sim \text{Categorical}(\pi_t)$.
8:     Query LLM $a_t$; observe $q_t$ and $c_t$ (only for $a_t$).
9:     $\tilde{c}_t \leftarrow \min(c_t/\tau, 1); \quad r_t \leftarrow w_t^q q_t - w_t^c \tilde{c}_t$.
10:    Compute batch baseline $b_t \leftarrow \frac{1}{B}\sum_{i=1}^{B} r^{(i)}$.
11:    $\mathcal{L}_t \leftarrow -(r_t - b_t) \log \pi_t[a_t] - \beta H(\pi_t)$.
12:    Update $\theta \leftarrow \theta - \eta \nabla_\theta \mathcal{L}_t$.
13: **end for**
14: **Inference (no retraining):** given $x$ and $w$, output $a^\star(x, w) = \arg\max_a \pi_\theta(a \mid x, w)$.

---

## 2.3 Objective and Learning Algorithm

Given the policy in equation 2 and reward in equation 1, the training objective is to find the parameters $\theta$ that maximize the expected cumulative reward:

$$\max_\theta \ J(\theta) = \mathbb{E}_{s_t \sim \mathcal{D}, a_t \sim \pi_\theta(\cdot \mid s_t)}\Big[ \sum_{t=1}^{T} r_t \Big]. \tag{4}$$

where the expectation is taken over the data distribution of contexts, $\mathcal{D}$, and the actions sampled from the policy. We optimize this objective using the REINFORCE policy gradient algorithm, enhanced with a baseline for variance reduction and entropy regularization for improved exploration. The per-sample loss function to be minimized is:

$$\mathcal{L}_t(\theta) = -(r_t - b_t) \log \pi_\theta(a_t \mid s_t) - \beta H(\pi_\theta(\cdot \mid s_t)), \tag{5}$$

where $H(\cdot)$ is the Shannon entropy of the policy distribution, $\beta \geq 0$ is a coefficient controlling the strength of the entropy regularization, and $b_t$ is a baseline used for variance reduction. We employ the mean reward over the current mini-batch as the baseline, defined as:

$$b_t = \frac{1}{B}\sum_{i=1}^{B} r_t^{(i)}, \tag{6}$$

where $B$ is the batch size and $r_t^{(i)}$ is the reward for the $i$-th example in the batch. While policy gradient methods are well-suited for training our policy, the formulation of our framework is general and can accommodate other classic learning algorithms, which we explore in our analysis in Sec. 4.6.

## 2.4 Training and Inference

**Training.** The policy network's parameters $\theta$ are optimized to maximize the expected reward using the REINFORCE algorithm detailed in Sec. 2.3. The training procedure has two key methodological features. First, to train a single policy that can serve diverse user preferences, we *randomly sample* the preference vector $w_t$ for each training instance (uniformly on the 1-simplex). Second, while our training utilizes pre-existing benchmark logs with complete information, we *simulate a bandit environment* to match deployment conditions. For each instance, after an action $a_t$ is sampled from the policy, the supervision signal is restricted to only the outcome of that specific action. The policy gradient updates are performed using the Adam optimizer (Kingma & Ba, 2017).

**Inference.** At deployment time, the router operates deterministically to exploit the learned policy. Given a prompt $x$ and a user-specified preference vector $w$, the router selects the action with the highest probability:

$$a^*(x, w) = \arg\max_{a \in A} \pi_\theta(a \mid x, w). \tag{7}$$

This allows operators to adjust the performance–cost behavior on a per-request basis by simply modifying the input vector $w$, without any need for retraining the model.

## 3 EXPERIMENTS SETUP

### 3.1 DATASETS AND BENCHMARKS

We evaluate on **RouterBench** (Hu et al., 2024) and two question-answering datasets (Kwiatkowski et al., 2019; Yang et al., 2018), which provide prompt-level logs with multiple candidate LLMs per query, including a task identifier, a performance score per LLM, and a monetary cost per LLM. While the benchmark logs contain scores/costs for *all* LLMs, our training strictly uses *bandit-consistent* supervision (only the chosen arm is observed).

Our experiments evaluate routing across a diverse set of widely used large language models, spanning both open-source and proprietary offerings. A detailed list and description of these models is provided in Appendix A.3.

**Tasks and Evaluation.** To evaluate our framework, we curate a set of eight distinct tasks(the dataset details are in A.4). Our model is trained on a mixture of data from five of these tasks: **GSM8K** (Cobbe et al., 2021), **MMLU** (Hendrycks et al., 2021), **ARC-C** (Clark et al., 2018), **Winogrande** (Sakaguchi et al., 2021), and **Natural Questions (NQ)** (Kwiatkowski et al., 2019). We create an 80%/20% training/testing split for each of these tasks and combine the training splits to form the full training set.

Our evaluation is then conducted in two settings:

- **In-Distribution Evaluation:** We test the model on the held-out 20% test sets of the five tasks it was trained on. This measures the model's ability to unseen examples from familiar tasks.
- **Out-of-Distribution Generalization:** To assess generalization to entirely new tasks, we evaluate the trained model on three benchmarks it has never seen during training: **MBPP** (Austin et al., 2021), **Hellaswag** (Zellers et al., 2019), and **HotpotQA** (Yang et al., 2018).

### 3.2 BASELINE METHODS

We compare our method against representative routers and common-sense baselines:

- **Smallest LLM** always routes to the smallest model.
- **Largest LLM** always routes to the largest model.
- **RouterDC** (Chen et al., 2024) learns dual-contrastive embeddings for queries and models, requires full-information labels.
- **GraphRouter** (Feng et al., 2025) learns graph-structured representations over queries, tasks, and models, also requires full labels.

### 3.3 METRICS

Following RouterBench (Hu et al., 2024), we evaluate methods on two axes:

- **Performance score** is a normalized value in $[0, 1]$ that indicates task success, derived either from exact match accuracy or from GPT-4 ratings for more open-ended tasks.
- **Monetary cost** is the estimated API call cost per query in USD.

### 3.4 IMPLEMENTATION DETAILS

Our policy is implemented in PyTorch. We use frozen all-MiniLM-L6-v2 (Wang et al., 2020) as the prompt encoder. The trainable components consist of two small MLPs with ReLU activations: one to encode the preference vector and a decision head that produces the final logits over the candidate LLMs. All prompts are tokenized to a maximum length of 512. We train our policy for 100 epochs using the Adam optimizer (Kingma & Ba, 2017) with a learning rate of $1 \times 10^{-4}$ and a batch size of 32. For the REINFORCE algorithm, we set the entropy regularization coefficient $\beta$ to 0.05. All experiments were conducted on NVIDIA A100 80GB GPUs.

Table 2: Testing score (%) on in-distribution tasks. The **best** results are highlighted in bold, and the second-best results are underlined.

| Methods | ARC-C | GSM8K | MMLU | Winogrande | NQ | Avg ↑ |
|---|---|---|---|---|---|---|
| Smallest LLM | 38.78 | 41.15 | 25.43 | 52.41 | 14.95 | 34.54 |
| Largest LLM | 96.19 | 65.88 | **81.19** | 81.93 | 29.15 | 70.87 |
| RouterDC | 91.99 | 59.68 | 60.98 | 74.74 | 31.00 | 63.68 |
| GraphRouter | 94.18 | **66.28** | 80.20 | 46.83 | 31.60 | 65.42 |
| Ours | **96.60** | 64.58 | 81.06 | **82.61** | **43.01** | **73.57** |

## 4  EXPERIMENTS RESULTS

### 4.1  PERFORMANCE ON IN-DISTRIBUTION TASKS

We first evaluate our method (BARP) against four baselines on in-distribution tasks, with results illustrated in Fig. 1 and reported in Table 2. BARP achieves the strongest trade-off between performance and cost. It delivers the highest average score (73.57%), outperforming the strong, full-information routers, RouterDC and GraphRouter, by a relative **15.53%** and **12.44%** respectively. It also establishes new best scores on ARC-C, Winogrande, and NQ. While the Largest LLM baseline is competitive on some tasks, its high monetary cost makes it impractical. In contrast, BARP achieves a performance level comparable to the strongest baselines while maintaining a cost significantly lower than other learned routers, establishing its superior efficiency on familiar tasks.

### 4.2  GENERALIZATION ABILITY TO NEW TASKS

To assess robustness, we further evaluate the trained models on out-of-distribution tasks they have never seen during training. As shown in Table 3, the full-information routers (RouterDC and GraphRouter) struggle to generalize, with their performance dropping sharply on MBPP and HpQA. In contrast, BARP demonstrates robust generalization, achieving the highest average score (66.08%) among all methods. It obtains the best score on HpQA, where other learned methods fail, and maintains performance competitive with the much more expensive Largest LLM baseline on MBPP and Hellaswag. This confirms that BARP preserves its superiority not only on in-distribution tasks but also when adapting to unseen tasks, confirming its robustness and practical deployment value.

Table 3: Testing score (%) on out-of-distribution tasks. The **best** results are highlighted in bold, and the second-best results are underlined.

| Methods | MBPP | Hellaswag | HpQA | Avg ↑ |
|---|---|---|---|---|
| Smallest LLM | 34.43 | 25.48 | 27.49 | 29.14 |
| Largest LLM | **68.62** | **83.96** | 40.93 | 64.50 |
| RouterDC | 39.06 | 69.60 | 25.00 | 44.55 |
| GraphRouter | 64.29 | 70.87 | 22.20 | 52.45 |
| Ours | 68.24 | 83.72 | **46.29** | **66.08** |

### 4.3  OVERALL PERFORMANCE AND COST-EFFECTIVENESS

Finally, to provide a holistic measure of performance and cost across all evaluation settings, we summarize the results by averaging across all eight tasks in Table 4. This view confirms that BARP provides the best balance of performance and cost. Compared to GraphRouter, the strongest offline baseline, our method improves the overall average score by **16.84%** while simultaneously reducing monetary cost by **50.00%**. In contrast, RouterDC provides a significant cost reduction but at the expense of a lower score, while the Largest LLM improves accuracy by 13.08% but at the expense of a more than threefold increase in cost. These results validate that our preference-conditioned, bandit-feedback approach is not only more effective but also substantially more cost-efficient than methods relying on full-information supervision.

Table 4: Comparison of methods in terms of Score, Cost, and the corresponding percentage Score improvements and Cost reduction rate, relative to the state-of-the-art method(GraphRouter (Feng et al., 2025)). The score and cost are averaged over in-distribution and out-of-distribution tasks. The cost is multiplied by $10^3$ for readability.

| Method | Score | Score Improvement (%) | Monetary Cost | Cost Reduction (%) |
|---|---|---|---|---|
| Smallest LLM | 32.52 | -46.30 | 0.05 | 94.68 |
| Largest LLM | 68.48 | 13.08 | 3.29 | -250.00 |
| RouterDC | 56.51 | -6.69 | 0.79 | 15.96 |
| GraphRouter | 60.56 | 0 | 0.94 | 0 |
| BARP (Ours) | 70.76 | 16.84 | 0.47 | 50.00 |

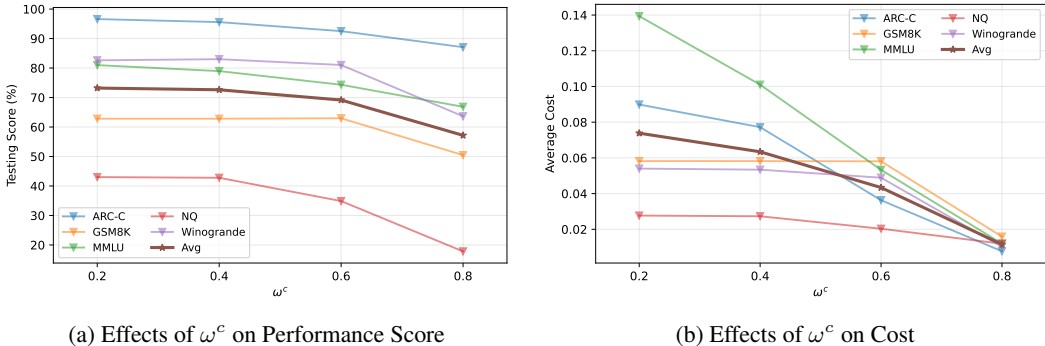

(a) Effects of $\omega^c$ on Performance Score  (b) Effects of $\omega^c$ on Cost

Figure 3: Effects of $\omega^c$

## 4.4 SENSITIVITY ANALYSIS

### 4.4.1 ANALYSIS OF THE PREFERENCE TRADE-OFF

We analyze the sensitivity of our router to the user-specified preference, which provides a direct trade-off between performance and cost. Recall from Sec. 2.1 that the preference vector is $w = (w^q, w^c)$, where $w^c$ is the weight on cost reduction. In this analysis, we vary the cost weight $w^c \in [0, 1]$ (with $w^q = 1 - w^c$) at inference time and observe its effect on the router's behavior. Figure 3 reports the effects of varying $w^c$ on both performance score and monetary cost across tasks.

As shown in Figure 3a, smaller values of the cost weight $w^c$ (e.g., 0.2) lead the router to prioritize performance, achieving strong scores across most tasks. For example, ARC-C remains above a 95% score and Winogrande above 80%. However, as $w^c$ increases, the average score gradually declines, most noticeably on NQ and MMLU, reflecting the router's increasing preference for cheaper models even when they are less performant.

Conversely, Figure 3b shows that larger $w^c$ values yield significant reductions in average cost. The cost decreases steadily from \$0.074 at $w^c = 0.2$ to only \$0.015 at $w^c = 0.8$, with consistent reductions across all tasks. This demonstrates that the router effectively adapts its selections in line with the user-specified trade-off, choosing lower-cost models when cost is emphasized.

Overall, these results confirm that the preference vector provides a clear and interpretable control knob for operators. Lower cost weights favor high performance at a higher cost, while higher cost weights sacrifice some performance to achieve substantial cost savings. This allows the behavior of BARP to be tuned to specific deployment requirements without any need for retraining.

### 4.4.2 IMPACT OF PROMPT ENCODER CHOICE

We analyze how the choice of the frozen prompt encoder affects routing performance. A more powerful encoder might provide better representations, but could also be less efficient. We compare three widely-used pre-trained models of increasing size: **all-MiniLM-L6-v2** (Wang et al., 2020) (384-dim), **BERT-base-uncased** (Devlin et al., 2018) (768-dim), and **E5-large-v2** (Wang et al.,

2022) (1024-dim). For each, we train only the preference encoder and the router's decision head using the same bandit-feedback procedure.

The results, averaged over all in-distribution tasks with a balanced preference ($w^q = w^c = 0.5$), are presented in Table 5. The all-MiniLM-L6-v2 encoder achieves the highest average score (0.7432), establishing the best trade-off between performance and model size. While the much larger E5-large-v2 performs comparably on score, its increased representational capacity does not translate into a significant routing advantage. Conversely, BERT-base-uncased yields a noticeably lower score, suggesting its representations are less effective for this task.

These findings provide a valuable insight: our routing framework does not require a large, resource-intensive model for prompt encoding. A compact, efficient sentence-level encoder like MiniLM is sufficient to capture the necessary semantics for routing. We hypothesize this is because modern sentence transformers, trained with contrastive objectives, produce more suitable sentence-level embeddings for this task than older models like BERT, which were trained on token-level objectives. Given its superior performance and smaller footprint, we use all-MiniLM-L6-v2 as the default encoder for all other experiments in this paper.

| Prompt Encoder | Avg Score | Avg Monetary Cost |
|---|---|---|
| MiniLM-L6-v2 | **0.7432** | 0.0007 |
| BERT-base-uncased | 0.7226 | **0.0005** |
| E5-large-v2 | 0.7418 | 0.0007 |

Table 5: Comparison of different frozen prompt encoders. Results are averaged across in-distribution tasks using a balanced preference ($w^q = w^c = 0.5$) during inference. The Avg Cost refers to the monetary cost of the LLMs selected by the router, not the encoder's cost.

## 4.5 IMPACT OF DECISION HEAD ARCHITECTURE

We also analyze the impact of the decision head's architecture, which sits atop the frozen encoder and maps the context representation to action logits. We evaluate three types of decision heads mentioned in Sec. 2.2: a simple **linear** layer, a parameter-efficient **bilinear** model, and a two-layer **MLP** with a ReLU non-linearity.

As shown in Table 6, the MLP head achieves the best overall performance, reaching the highest average score (0.7432). The linear head is competitive, suggesting that a direct mapping is a strong baseline, while the bilinear head underperforms. These results provide a key insight: while a simple linear mapping is effective, the added representational capacity of the MLP's non-linearity is beneficial for learning the complex function that maps a prompt and a user preference to the optimal LLM choice.

| Head Type | Avg Score | Avg Monetary Cost |
|---|---|---|
| Linear | 0.7396 | 0.0007 |
| Bilinear | 0.7317 | **0.0006** |
| MLP | **0.7432** | 0.0007 |

Table 6: Comparison of different decision head architectures. Results are averaged across in-distribution tasks, using a balanced preference ($w^q = w^c = 0.5$) during inference.

We hypothesize that the bilinear head, despite being designed to model interactions, may be more difficult to optimize with the sparse signal provided by bandit feedback, potentially leading to its lower score. Given that the MLP head provides the best performance without a significant increase in complexity, we adopt it as the default architecture for all other experiments.

## 4.6 ANALYSIS OF LEARNING ALGORITHMS

A key feature of our framework is its flexibility to accommodate different learning algorithms. To analyze the impact of the algorithm choice, we compare our policy-gradient approach (REINFORCE) with several classic contextual bandit strategies: **Linear Thompson Sampling (LinTS)** (Agrawal & Goyal, 2013), **LinUCB** (Li et al., 2010), and $\epsilon$-**greedy**. To ensure a fair comparison, all algorithms operate on the identical context representation (the concatenated prompt and preference embeddings). As is standard, the classic bandit strategies are paired with a linear model to map these features to rewards, while our main approach uses a non-linear MLP.

Table 7 presents the results evaluated with a balanced preference ($w^q = w^c = 0.5$). The policy-gradient method (REINFORCE) achieves a substantially higher average score, demonstrating supe-

rior performance on this task. Notably, bandit approaches tend to yield slightly lower costs, suggesting that their conservative exploration might favor cheaper models at the expense of performance.

The primary finding from this analysis is that the routing decision function is inherently complex. While classic bandit algorithms provide a strong baseline, their performance is limited by the linear assumptions they make about the relationship between context and reward. The significant performance gap suggests that an algorithm capable of learning a non-linear policy, such as REINFORCE paired with an MLP, is necessary to effectively model the nuances of LLM routing.

| Method | Avg Score | Avg Monetary Cost |
|---|---|---|
| LinTS | 0.6430 | 0.00046 |
| LinUCB | 0.6166 | **0.00044** |
| $\epsilon$-greedy | 0.6556 | 0.00056 |
| REINFORCE | **0.7432** | 0.00070 |

Table 7: Comparison between REINFORCE and classical bandit algorithms. Results are averaged across in-distribution tasks, using a balanced preference ($w^q = w^c = 0.5$) during inference.

## 5 ADDITIONAL RELATED WORK

**LLM routing.** With the rapid growth of LLMs, there is increasing interest in routing strategies that decide which model to query for each input. Early approaches often rely on ensembles, such as majority voting over all outputs, or static heuristics like always choosing the largest or smallest model. Recently, learning-based routers have been proposed. GraphRouter (Feng et al., 2025) learns graph-structured representations across prompts, tasks, and models to exploit relational information. RouterDC (Chen et al., 2024) introduces dual-contrastive objectives for aligning query and model embeddings. Other efforts design mixture-of-experts systems that dynamically allocate queries across LLMs (Varangot-Reille et al., 2025).

**Contextual bandits.** The contextual bandit framework (Langford & Zhang, 2007) formalizes decision-making under partial feedback: at each round, the learner observes a context, selects an action, and only receives feedback for that action. Classical bandit algorithms include LinUCB (Li et al., 2010), which uses optimism in linear reward models; Thompson Sampling (Agrawal & Goyal, 2013), which maintains a posterior over reward parameters; and $\epsilon$-greedy strategies, which trade off exploration and exploitation through randomization. Beyond linear settings, neural contextual bandits extend these ideas with non-linear function approximators (Riquelme et al., 2018; Zhou et al., 2020). Bandit methods have been applied to recommendation (Li et al., 2010), online advertising (Chapelle & Li, 2011), and adaptive experiment design.

## 6 CONCLUSION AND DISCUSSION

In this work, we address the challenge of efficiently selecting the optimal LLM from a pool of candidates to balance performance and cost. We formalize this task as a preference-conditioned contextual bandit problem and introduce BARP. Trained with policy gradients on bandit feedback, our method learns to map a user's prompt and specific performance-cost preference to the most suitable LLM. Extensive experiments demonstrate that BARP significantly outperforms both top-performing individual LLMs and strong offline routers on both in-distribution and out-of-distribution tasks. Crucially, we show that the preference vector provides an effective and interpretable control mechanism, allowing operators to tune the router's behavior at inference time without retraining.

We acknowledge several limitations for future improvement. Our method trains on static, offline logs, which is practical but differs from a truly online setting where a router could learn continuously from live feedback. We only consider performance and monetary cost, while real deployments may require richer, possibly task-specific preferences and constraints (e.g., latency). The current contextual bandit formulation also models routing as a single-step decision, making it well-suited for many tasks but not explicitly designed for multi-turn, conversational scenarios. Furthermore, our experiments focused on a pool of general-purpose LLMs, and future work could explore routing to highly specialized, domain-expert models.

## ETHICS STATEMENT

The primary goal of this research is to improve the efficiency of using large language models, a direction with a positive societal impact. By enabling users to select smaller, less expensive models when appropriate without a significant loss in performance, our work contributes to reducing the overall energy consumption and carbon footprint associated with deploying these powerful but resource-intensive technologies. Our work relies on existing, publicly available benchmark datasets and pre-trained language models. We do not use any private or personally identifiable information, and our research does not involve human subjects. As with any system that improves the efficiency of LLM routing, there is a possibility of misuse, for example, in routing to optimize spam or misinformation generation. However, we believe the risk is limited and outweighed by the benefits of more efficient LLM routing.

## REPRODUCIBILITY STATEMENT

We are committed to ensuring the reproducibility of our work. To this end, all code required to replicate our experiments, including scripts for training, evaluation, and all analyses presented in the paper, will be made publicly available upon publication in an open-source repository.

**Datasets.** Our primary experiments are conducted on the publicly available benchmarks. We will provide scripts to download and process all data into the format required by our codebase. Our data splits are deterministic, based on the random seed provided in our code.

**Models and Hyperparameters.** The specific pre-trained models used for the prompt encoder and the full list of candidate LLMs are detailed in the appendix. All critical hyperparameters, including learning rates, batch sizes, and regularization coefficients, are reported in 3.4. Our code is implemented in PyTorch.

**Computational Resources.** All experiments were conducted on a single NVIDIA A100 GPU with 80GB of memory. The training for our main model completes in approximately 2-3 hours. The code for the classic bandit baselines is also provided and runs efficiently on a standard CPU.

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

# A APPENDIX

## A.1 NOTATION

Table 8: Summary of notations.

| Symbol | Description |
|--------|-------------|
| *Problem Formulation* | |
| $K$ | Total number of candidate LLMs (actions). |
| $A$ | The set of actions $\{1, \ldots, K\}$. |
| $t$ | The time step or round index. |
| $x_t$ | The input prompt at round $t$. |
| $w_t$ | The user preference vector $(w_t^q, w_t^c)$ at round $t$. |
| $w_t^q, w_t^c$ | The weights for performance score and cost, respectively. |
| $s_t$ | The context (state) at round $t$, defined as the tuple $(x_t, w_t)$. |
| $a_t$ | The action (chosen LLM) at round $t$. |
| $q_t$ | The performance score of the chosen LLM's output, $q_t \in [0, 1]$. |
| $c_t$ | The monetary cost of using the chosen LLM. |
| $\tilde{c}_t$ | The normalized monetary cost, $\min(c_t/\tau, 1)$. |
| $r_t$ | The scalar reward at round $t$. |
| $\mathcal{D}$ | The underlying data distribution of contexts. |
| *Policy and Learning* | |
| $\theta$ | The trainable parameters of the policy network. |
| $\pi_\theta(a\|s)$ | The policy; probability of selecting action $a$ given context $s$. |
| $h(\cdot)$ | The frozen prompt encoder function. |
| $\phi(\cdot)$ | The preference encoder (MLP) function. |
| $z$ | The concatenated context representation $[h(x); \phi(w)]$. |
| $g_\theta(\cdot)$ | The decision head of the policy network. |
| $o$ | The vector of logits produced by the decision head. |
| $a^*$ | The optimal action selected at inference time (via argmax). |
| $J(\theta)$ | The expected cumulative reward objective function. |
| $\mathcal{L}_t(\theta)$ | The policy gradient loss function at round $t$. |
| $b_t$ | The reward baseline (batch-mean reward). |
| $B$ | The batch size used during training. |
| $H(\cdot)$ | The Shannon entropy function. |
| $\beta$ | The entropy regularization coefficient. |
| $\tau$ | The cost scaling and capping hyperparameter. |

## A.2 ADDITIONAL RESULTS

Table 9: Testing score (%) of each candidate LLM on in-distribution tasks.

| Candidate LLM | ARC-C | GSM8K | MMLU | Winogrande | Avg ↑ |
|---------------|-------|-------|------|------------|-------|
| WizardLM/WizardLM-13B-V1.2 | 61.02 | 50.63 | 44.65 | 50.75 | 51.76 |
| claude-instant-v1 | 80.27 | 62.72 | 59.64 | 61.96 | 66.15 |
| claude-v1 | 86.87 | 65.08 | 65.72 | 65.98 | 70.91 |
| claude-v2 | 86.87 | 66.26 | 62.81 | 66.06 | 70.50 |
| gpt-3.5-turbo-1106 | 83.06 | 60.48 | 64.71 | 57.93 | 66.55 |
| gpt-4-1106-preview | 96.19 | 65.88 | 81.19 | 81.93 | 81.30 |
| meta/code-llama-instruct-34b-chat | 37.35 | 45.66 | 0.48 | 38.44 | 30.48 |
| meta/llama-2-70b-chat | 73.40 | 52.30 | 2.68 | 48.22 | 44.15 |
| mistralai/mistral-7b-chat | 38.78 | 41.15 | 25.43 | 52.41 | 39.44 |
| mistralai/mixtral-8x7b-chat | 83.20 | 51.90 | 63.51 | 55.25 | 63.47 |
| zero-one-ai/Yi-34B-Chat | 86.12 | 54.81 | 65.85 | 62.90 | 67.42 |

Table 10: Testing score (%) of each candidate LLM on out-of-distribution tasks.

| Candidate LLM | MBPP | Hellaswag | Avg ↑ |
|---|---|---|---|
| WizardLM/WizardLM-13B-V1.2 | 37.00 | 33.38 | 35.19 |
| claude-instant-v1 | 60.42 | 58.51 | 59.47 |
| claude-v1 | 59.72 | 56.85 | 58.29 |
| claude-v2 | 64.17 | 62.42 | 63.30 |
| gpt-3.5-turbo-1106 | 65.34 | 58.66 | 62.00 |
| gpt-4-1106-preview | 68.62 | 83.96 | 76.29 |
| meta/code-llama-instruct-34b-chat | 51.76 | 20.82 | 36.29 |
| meta/llama-2-70b-chat | 33.02 | 52.59 | 42.81 |
| mistralai/mistral-7b-chat | 34.43 | 25.48 | 29.96 |
| mistralai/mixtral-8x7b-chat | 54.10 | 41.69 | 47.90 |
| zero-one-ai/Yi-34B-Chat | 38.64 | 74.26 | 56.45 |

## A.3 CANDIDATE LLMS

For tasks from RouterBench (Hu et al., 2024), we have candidate LLMs as follows: (i) **WizardLM-13B-V1.2** (Xu et al., 2023) is a fine-tuned instruction-following model from the WizardLM series; (ii) **Claude-instant-v1** is a lightweight model from Anthropic optimized for speed; (iii) **Claude-v1** is Anthropic's first-generation flagship model; (iv) **Claude-v2** (Anthropic, 2023) is an improved successor with stronger reasoning ability; (v) **GPT-3.5-turbo-1106** is OpenAI's production-grade model designed for efficiency and broad coverage; (vi) **GPT-4-1106-preview** (OpenAI et al., 2023) is OpenAI's most capable general-purpose model at the time of release; (vii) **Code Llama Instruct-34B-Chat** (Rozière et al., 2024) is a code-specialized instruction-tuned model; (viii) **Llama-2-70B-Chat** (Touvron et al., 2023) is a general conversational model trained with reinforcement learning from human feedback; (ix) **Mistral-7B-Chat** (Jiang et al., 2023) is an efficient chat-optimized model from Mistral AI; (x) **Mixtral-8x7B-Chat** (Jiang et al., 2024) is Mistral's mixture-of-experts model offering higher throughput; and (xi) **Yi-34B-Chat** (Young et al., 2024) is a large-scale bilingual chat model with strong performance in both English and Chinese.

For NQ and HpQA datasets, the candidate LLMs consist of Llama-3.1-8b-instruct (Grattafiori et al., 2024), Llama-3.1-70b-instruct (Grattafiori et al., 2024)2, mistral-7b-instruct-v0.3 (Jiang et al., 2023), qwen2.5-7b-instruct (Yang et al., 2024), gemma-2-27b-it (Team et al., 2024), mixtral-8x22b-instruct-v0.1 (Jiang et al., 2024).

## A.4 DATASET DETAILS

- **GSM8K** (Cobbe et al., 2021): A dataset of diverse grade school math word problems, testing a model's ability to perform multi-step mathematical reasoning.

- **MMLU** (Hendrycks et al., 2021): A benchmark that measures the knowledge acquired by models during pretraining and evaluates models in zero-shot and few-shot settings across 57 tasks, testing both knowledge and reasoning on different fields of human knowledge.

- **ARC-C** (Clark et al., 2018): A rigorous question answering dataset, ARC-Challenge includes complex, different grade-school level questions that require reasoning beyond simple retrieval, testing the true comprehension capabilities of models. Arc Challenge dataset contains those that both a retrieval and a co-occurrence method fail to answer correctly)

- **Winogrande** (Sakaguchi et al., 2021): A large-scale and increased harness dataset inspired by the original Winograd Schema Challenge(WSC) tests models on their ability to resolve pronoun ambiguity and their ability to understand the context with commonsense knowledge.

- **NQ** (Kwiatkowski et al., 2019): A comprehensive collection of real user queries submitted to Google Search, with answers sourced from Wikipedia by expert annotators.

- **MBPP** (Austin et al., 2021): The benchmark is designed to be solvable by entry-level programmers, covering programming fundamentals, standard library functionality, etc. Each problem comprises a task description, code solution, and 3 automated test cases.

- **Hellaswag** (Zellers et al., 2019): This dataset challenges models to pick the best ending choice for a given sentence. It uses Adversarial Filtering(AF) to create a Goldilocks zone of complexity, wherein generations are largely nonsensical to humans but always make models struggle.
- **HpQA** (Yang et al., 2018): This dataset is designed for question answering and features natural, multi-hop questions. It provides strong supervision for supporting facts, enabling the development of more explainable question answering systems.

### A.5 USE OF LLMS

The LLM's role was strictly a writing and editing assistant, used to augment and refine the work.

The primary uses of the LLM included:

- **Refining Prose and Tone:** Improving the clarity, flow, and academic tone of sentences and paragraphs across all sections.
- **Ensuring Consistency:** Cross-referencing the manuscript to identify and correct inconsistencies in terminology, notation, and quantitative claims between the text and tables.

All scientific contributions, including the core ideas, experimental design, analysis, and final claims, were conceived and executed by the authors. The LLM served as a tool to help articulate these contributions more effectively.

