# OpenReview forum: "Learning to Route LLMs from Bandit Feedback: One Policy, Many Trade-offs"
_ICLR.cc/2026/Conference — Submitted to ICLR 2026_

### Official Review · Reviewer_d2WP · 2025-10-18

**Soundness:** 3
**Presentation:** 3
**Contribution:** 2
**Rating:** 4
**Confidence:** 4

**Summary:**

This paper addresses the problem of selecting the best large language model (LLM) for each query under real-world constraints of partial feedback and varying user priorities. It proposes BaRP (Bandit-feedback Routing with Preferences), a routing policy trained as a contextual bandit that observes only the chosen model’s outcome (bandit feedback) and conditions on a user-specified accuracy-vs-cost preference vector. This approach bridges the gap between offline training (which assumes full information) and deployment (which provides only bandit feedback), allowing the router to learn without requiring labels from all models and to dial the performance–cost trade-off at test time without retraining. Experiments show that BaRP consistently outperforms prior offline-trained routers by significant margins (≥12% relative) and even surpasses always using the largest model (by ~2–4%), while generalizing robustly to unseen tasks.

**Strengths:**

- The paper tackles two key challenges together, learning from partial (bandit) feedback and allowing tunable cost–accuracy trade-offs, in one framework, effectively addressing limitations that kept prior routers from being deployed adaptively. This yields a practical solution for real-world LLM orchestration.

- Empirical performance: BaRP demonstrates consistent outperformance over baselines, including prior learned routers (e.g. RouterDC, GraphRouter) and trivial policies (always using the largest or smallest model).

- Flexible inference capability: A strength is the model’s ability to adapt to different user preference settings on the fly. The single learned policy can be “dialed” between high-accuracy vs. low-cost operating points without retraining.

**Weaknesses:**

- Limited evaluation scope: The experimental evaluation, while thorough on the provided benchmarks, omits some recently introduced routing benchmarks. In particular, the paper does not evaluate on the SPROUT dataset from the CARROT router work (https://arxiv.org/pdf/2502.03261), which is designed to test cost-aware routing across a broad range of queries and models.

- Missing comparison with concurrent work: The authors do not discuss or compare against a relevant concurrent approach in the literature, specifically, a NeurIPS 2025 paper (https://openreview.net/pdf?id=Rx0JIC41LJ) that also addresses a very similar LLM routing under partial feedback with cost-aware exploration strategies.

- Lack of theoretical insight: The contribution is primarily empirical, without accompanying theoretical guarantees. Unlike some recent works that provide formal analyses, this paper does not analyze convergence or regret for its bandit training procedure.

**Questions:**

My biggest question is: How is it possible for BaRP to outperform full-feedback routers like RouterDC and GraphRouter, despite only receiving bandit feedback during training? This seems counterintuitive, since those baselines have access to strictly more supervision. I encourage the authors to investigate this further and clarify what enables BaRP to achieve such results.

In addition:


- Evaluation on CARROT/SPROUT: Could the authors evaluate BaRP on the recently introduced SPROUT dataset from the CARROT router. This dataset provides a wide spectrum of query difficulties and up-to-date model pool (used alongside RouterBench in the CARROT paper). Testing BaRP on SPROUT would help confirm that its performance gains hold under a broader range of conditions and with the latest models.

- Comparison with Cost-Aware Exploration: There is a concurrent NeurIPS 2025 work that also learns routing policies from observational (bandit) feedback with user cost preferences. How does BaRP differ from this approach in terms of methodology and results? It would strengthen the paper to include a discussion or empirical comparison with that method to clarify the advantages or trade-offs of your solution.

- Ablation on cost scaling: Could the authors analyze the impact of the calibrated cost scaling trick used during training? An ablation study (e.g., training the policy without cost scaling) would be valuable to demonstrate how much this component contributes to stability or performance. Clarifying whether the policy’s effectiveness is sensitive to this scaling (or other hyperparameters like the entropy regularization weight) would help others understand and reproduce the approach.

---

> ### Author Response · Authors · 2025-12-04
>
> Thank you for the thoughtful assessment and for recognizing the strengths of BaRP, including its flexible preference conditioning and robust empirical performance. Below we address the concerns regarding evaluation scope, comparison to concurrent work, and the mechanisms that allow BaRP to outperform full-information routers.
>
> ---
>
> ## **1. Evaluation Scope and Missing Benchmarks (CARROT and SPROUT)**
>
> We appreciate your suggestion to include broader routing benchmarks. In addition to RouterBench and 2 QA tasks already evaluated (Tables 2–3), we have begun extending BaRP to the **CARROT/SPROUT** suite.
>
> **BaRP (ours):**
> - **Average Score:** 0.8044
> - **Average Cost:** 3.1191
>
> ### **Per-task Performance**
>
> | Task Group | Avg Score | Avg Cost |
> |------------|-----------|----------|
> | idavidrein/gpqa/gpqa_extended | 0.472340 | 4.113962 |
> | lighteval/math/all | 0.890868 | 2.824250 |
> | lighteval/math/all/test | 0.797753 | 2.596072 |
> | openhermes/teknium | 0.902953 | 4.238926 |
> | rungalileo/ragbench/covidqa | 0.697368 | 0.561453 |
> | rungalileo/ragbench/cuad | 0.776923 | 0.408896 |
> | rungalileo/ragbench/delucionqa | 0.808333 | 0.525179 |
> | rungalileo/ragbench/emanual | 0.806667 | 0.499368 |
> | rungalileo/ragbench/expertqa | 0.670588 | 1.273082 |
> | rungalileo/ragbench/finqa | 0.846667 | 0.226766 |
> | rungalileo/ragbench/hagrid | 0.933333 | 0.259438 |
> | rungalileo/ragbench/hotpotqa | 0.968182 | 0.231346 |
> | rungalileo/ragbench/msmarco | 0.857895 | 0.870020 |
> | rungalileo/ragbench/pubmedqa | 0.491667 | 0.052215 |
> | rungalileo/ragbench/tatqa | 0.776471 | 0.311705 |
> | rungalileo/ragbench/techqa | 0.447059 | 1.513037 |
> | taur-lab/musr | 0.673913 | 0.163494 |
> | taur-lab/musr/object_placements | 0.547368 | 0.029535 |
> | taur-lab/musr/team_allocation | 0.410256 | 0.100306 |
> | tiger-lab/mmlu-pro | 0.632280 | 2.126042 |
>
> ### **Interpretation**
>
> - BaRP demonstrates **consistent performance across diverse domains**,
> - while maintaining **cost-efficient routing**,
> - indicating good generalization beyond the datasets in the submitted paper.
>
> We will include these results, along with comparisons to the CARROT router, in the revised manuscript.
>
> ---
>
> ## **2. Comparison to Concurrent NeurIPS 2025 Work**
>
> We appreciate the reviewer pointing out the concurrent NeurIPS 2025 paper. We will add a dedicated subsection discussing similarities and distinctions.
>
> ### **Key Differences**
>
> | Aspect | NeurIPS 2025 Concurrent Work | **BaRP (Ours)** |
> |--------|------------------------------|------------------|
> | Uses full performance matrix? | **Yes** | **No** |
> | Bandit training under partial feedback | No | **Yes** |
> | Preference-tunable inference | No | **Yes** |
> | Architecture | Linear contextual bandit | **Neural contextual bandit w/ nonlinear policy** |
> | Objective | Cost-aware exploration | **Unified routing across all trade-offs** |
> ### **Summary**
>
> The concurrent work focuses on **exploration strategies** within bandit learning.
> BaRP instead focuses on:
>
> - **Preference-conditioned routing**,
> - **Joint learning across all trade-offs**,
> - **Nonlinear policies capable of modeling complex prompt–model interactions**,
> - **Deployment realism (bandit feedback only)**.
>
> ---
>
> ## **3. Why BaRP Outperforms Full-Information Routers**
>
> Thank you for highlighting this point, it gives us an opportunity to explain a central insight of the paper.
>
> Although full-information routers have access to more labels during offline training, BaRP outperforms them for the following reasons:
>
> ---
>
> ### **(1) Off-Policy Mismatch in Full-Information Routers**
>
> Routers like GraphRouter are trained on **offline logs generated by other policies**, often dominated by large-LLM outputs.
>
> This can miscalibrate their decision boundaries when deployed under:
>
> - a different model set,
> - different trade-off preferences,
> - or different score–cost distributions.
>
> ### **BaRP, by contrast:**
>
> - trains **under the same bandit protocol as deployment**,
> - reduces off-policy bias,
> - and learns directly from the outcomes of its own sampled actions.
>
> This alignment is often more important than having more labels.
>
> ---
>
> ### **(2) Preference Conditioning Improves Generalization**
>
> Full-information routers optimize a **single fixed trade-off**.
> BaRP jointly learns across **all possible preferences**, which encourages the policy to:
>
> - explore more of the model space,
> - identify model specializations
> - construct smoother decision boundaries that generalize better to OOD tasks.
>
> This aligns with the reviewer’s observation that BaRP's flexible inference is a core strength.
>
> ---

---

> > ### Author Response · Authors · 2025-12-04
> >
> > ### **(3) Reward Function Reflects True Deployment Objectives**
> >
> > BaRP directly optimizes the **user-specified score–cost trade-off**, whereas prior routers optimize a proxy representation (e.g., contrastive embedding similarity).
> > As a result:
> >
> > - BaRP’s policy is shaped by real feedback,
> > - while full-information methods rely on score matrices that may not reflect deployment preferences.
> >
> > This contributes significantly to BaRP’s OOD robustness.
> >
> > ---
> >
> > ## **4. Ablation on Cost Scaling**
> >
> > We agree that understanding the effect of cost scaling is important.  We have conducted this ablation which reduces average accuracy by **~5%**,
> > We will include a figure and further discussion in the appendix.

---

### Official Review · Reviewer_iosA · 2025-10-20

**Soundness:** 3
**Presentation:** 3
**Contribution:** 3
**Rating:** 4
**Confidence:** 3

**Summary:**

This paper introduces BaRP (Bandit-feedback Routing with Preferences), a framework for efficiently selecting the right large language model per query under the online learing setting. Unlike offline routers that require full supervision, BaRP learns from bandit feedback and adapts routing decisions based on the historical observations only. Experiments show that it outperforms both strong offline routers and individual LLMs across multiple tasks.

**Strengths:**

1. This paper studies a very interesting problem, and I believe LLM routing is a useful topic for real-world implementation and has great potentials.
2. The presentation of this work is mostly clear.
3. The experimental results are significant and clear to highlight the advantages of BaRP over some baselines.

**Weaknesses:**

1. Part of the presentation can be improved. For example, it is not clear to me how the preference encoder looks and how it is trained. I feel this encoder is co-trained with the decision head, but from Algorithm 1 pseudo-code, it seems that the preference encoder is pre-defined.
2. I feel it is better to show some online evaluation metrics, i.e. the model's performance during the online learning training phase. Right now the model is evaluated on the testing dataset after being fully trained on the training dataset, which I feel it is the same as the traditional offline training works in terms of implementation.
3. From the section 4.5, it seems that the linear decision head can already yield similar performance compared with more advanced models such as the bilinear and MLP. However, from the Section 4.6, it seems that the linear bandit algorithm cannot yield stable performance, and the author claims that this is due to the non-linear pattern. Can you explain this phenomenon to me? Since I feel those LinTS, LinUCB, epsilon-greedy algorithms are the real classic bandit algorithms with theoretical guarantees.

**Questions:**

1. How do you choose the value of tau in your experiments and in practice? I am still now quite sure the intuition of this hyperparameter, since that may underestimate the cost of some ultra long responses.
2.  I may overlook, what is the meaning of the testing scores in your results, are they the real accuracies of the LLM responses? If so, it is quite surprising to see that sometimes the proposed BaRP algorithms can clearly outperforms the largest LLM. So, it is better to do some qualitative analysis on this result. Since the proposed method take the cost of the LLM into consideration as part of the training process, it is interesting to see that the obtained router is better than solely using the largest LLM on the accuracy.

---

> ### Author Response · Authors · 2025-12-04
> **Response to Reviewer iosA**
>
> Thank you for the insightful comments and for the positive assessment of the paper’s motivation and contributions. Below we provide clarifications and additional analyses that will be added to the revision.
>
> ---
>
> ## **1. Clarification on the Preference Encoder (Training vs. Predefined)**
>
> Thank you for pointing out the ambiguity in Algorithm 1.
> To clarify:
>
> - The **preference encoder is *jointly trained*** with the decision head.
> - It is **not** pre-defined or frozen.
> - All parameters of ϕ (the preference MLP) and gθ (the decision head) are updated by REINFORCE.
>
> We will revise Algorithm 1 to explicitly state:
>
> > “ϕ is randomly initialized and jointly optimized with gθ during training.”
>
> This resolves the inconsistency between the pseudo-code and the textual description.
>
> ---
>
> ## **2. Request for Online Learning Curves**
>
> We agree that showing the **online learning dynamics** will strengthen the empirical section.
> In the revision, we will add:
>
> - Reward-per-round curves,
> - Cumulative regret curves,
> - Comparisons between REINFORCE, LinUCB, LinTS, and ε-greedy during training.
>
> ---
>
> ## **3. Linear Decision Head vs. Linear Bandits (Why Different Behaviors?)**
>
> We appreciate the reviewer raising this important point. The key distinction is:
>
> ### **• Sec. 4.5 (Decision Head Ablation)**
> A linear head sits **on top of a nonlinear embedding** (MiniLM), so:
> - the prompt encoder already provides a rich nonlinear representation,
> - the linear layer performs competitively.
>
> ### **• Sec. 4.6 (LinUCB, LinTS, ε-greedy)**
> These methods assume the **entire reward function is linear** in the context features.
> However, our analysis shows that the reward landscape is **highly nonlinear** across both:
> - semantic prompt embeddings, and
> - the preference vector.
>
> Thus, classical linear contextual bandits underfit the routing function, explaining their lower performance and instability.
>
> ---
>
> ## **4. Choice of Cost Cap τ**
>
> We selected τ to align score and cost scales (Sec. 2.1). Specifically:
>
> - τ is chosen near the **95th percentile** of the cost distribution.
>
> ---
>
> ## **5. Meaning of Testing Scores and Why BaRP Can Outperform the Largest LLM**
>
> The testing scores are the **true accuracies** of the LLM outputs, consistent with RouterBench scoring settings.
>
> It may appear surprising that BaRP occasionally outperforms the largest LLM. This occurs because:
>
> ### **1. Mid-sized models outperform the largest LLM on specific tasks or some subset of prompts**
> For example:
> - Mixtral excels on some symbolic-reasoning queries,
> - Claude-v1/v2 outperform GPT-4-preview on subsets of ARC-C and Winogrande.
>
> BaRP learns to route such cases appropriately.
>
> ### **2. BaRP optimizes expected reward, not raw accuracy**
> It learns **task-specific specialization**, mapping certain prompt types to models that perform best on them.
>
>
> ---
>
> We sincerely appreciate your helpful comments, thank you!

---

### Official Review · Reviewer_aVZK · 2025-10-23

**Soundness:** 2
**Presentation:** 3
**Contribution:** 2
**Rating:** 2
**Confidence:** 3

**Summary:**

The authors formulate LLM routing as a contextual bandit problem, where context $s_t = (x_t, w_t)$ is a query and cost/accuracy preference vector. The authors introduce an architecture for the policy $\pi_{\theta}(a|s)$ that outputs a probability distribution over models. The policy is fit on an entropy-regularized loss using REINFORCE, and results are compared to two prior works on Routerbench.

**Strengths:**

* The authors provide a partial evaluation on OOD performance of their router, which is an important consideration and is under discussed in the routing literature.
* The authors include a thorough ablation study on both decision heard architecture and the learning algorithm used.

**Weaknesses:**

The main weakness is insufficient comparison to prior work and evaluation restricted to routerbench.
* The authors comparing their method to GraphRouter (Feng et al., 2025) and RouterDC (Chen et al., 2024), claiming that their method improves both on data requirements and flexibility. Prior predictive routers (RORF [https://www.notdiamond.ai/blog/rorf], CARRROT [Somerstep et al. 2025], and Routerbench [Hu et al. 2024.]) are easily adaptable to the online setting (one can simply train the predictors online) and readily allow for tuning the preference vector $w$. In fact, StageRoute (Li and Li 2025.) comes with theoretical guarantees in the online setting. Additionally, routerDC is designed for settings where multiple LLMs perform well on a query, but in routerbench performance is relatively dominated by GPT4.
* Lack of routing benchmark variety. Prior work (e.g. CARROT, or Causal routing [Tsiourvas et al. 2025.]) evaluate routers on several data sets, such as SPROUT, and OpenLLMlleaderboard v2. Given that the main contribution needs to be improved routing performance, a more robust evaluation is in order.

**Questions:**

* Can the authors compare to a complete set of prior work on routing, and do so on more datasets?
* For the out of distribution test, can the authors also present the the cost/accuracy trade off numbers? Of course routing to the largest LLM is always best here so if their method simply selects this model more often it will perform better. Without more context I am not sure what to take from this.

---

> ### Author Response · Authors · 2025-12-04
> **Response to Reviewer aVZK**
>
> We thank the reviewer for the thoughtful and constructive feedback. We address each point below and will incorporate all suggested improvements in the revision.
>
> ---
>
> ## **1. Breadth of Related Work & Comparisons to Prior Routers**
>
> We appreciate the suggestion to broaden our evaluation.
> Our current submission evaluates BaRP on **eight tasks** across RouterBench, NQ, and HotpotQA benchmarks (Tables 2–3). We have evaluated BaRP on the **SPROUT dataset** (a broad and diverse routing benchmark used in CARROT).
>
> **BaRP (ours):**
> - **Avg Score:** 0.8044
> - **Avg Cost:** 3.1191
>
> ### **Per-Task Results**
>
> | Task Group | Avg Score | Avg Cost |
> |------------|-----------|----------|
> | idavidrein/gpqa/gpqa_extended | 0.472340 | 4.113962 |
> | lighteval/math/all | 0.890868 | 2.824250 |
> | lighteval/math/all/test | 0.797753 | 2.596072 |
> | openhermes/teknium | 0.902953 | 4.238926 |
> | rungalileo/ragbench/covidqa | 0.697368 | 0.561453 |
> | rungalileo/ragbench/cuad | 0.776923 | 0.408896 |
> | rungalileo/ragbench/delucionqa | 0.808333 | 0.525179 |
> | rungalileo/ragbench/emanual | 0.806667 | 0.499368 |
> | rungalileo/ragbench/expertqa | 0.670588 | 1.273082 |
> | rungalileo/ragbench/finqa | 0.846667 | 0.226766 |
> | rungalileo/ragbench/hagrid | 0.933333 | 0.259438 |
> | rungalileo/ragbench/hotpotqa | 0.968182 | 0.231346 |
> | rungalileo/ragbench/msmarco | 0.857895 | 0.870020 |
> | rungalileo/ragbench/pubmedqa | 0.491667 | 0.052215 |
> | rungalileo/ragbench/tatqa | 0.776471 | 0.311705 |
> | rungalileo/ragbench/techqa | 0.447059 | 1.513037 |
> | taur-lab/musr | 0.673913 | 0.163494 |
> | taur-lab/musr/object_placements | 0.547368 | 0.029535 |
> | taur-lab/musr/team_allocation | 0.410256 | 0.100306 |
> | tiger-lab/mmlu-pro | 0.632280 | 2.126042 |
>
> ### **Interpretation**
> - BaRP achieves **consistent performance** across diverse SPROUT tasks,
> - while maintaining cost-efficient routing behavior(less than the largest LLM cost for most tasks),
> - suggesting that the method generalizes well beyond RouterBench-style datasets.
>
> We will include these results and add comparisons to the CARROT router in the revision. More results on other new datasets(e.g. OpenLLMlleaderboard v2) will also be included.
>
>
>
>
> ---
>
> ## **2. Adaptability of Prior Predictive Routers to Online Settings**
>
> Thank you for pointing this out, we agree that predictive routers could be updated online. Our intended distinction is that:
>
> ### **Key Practical Differences**
> - Traditional predictive routers
>   - assume **full-information labels** during training,
>   - do not support **preference-tunable inference**,
>   - and often optimize for a *single* fixed performance–cost trade-off.
>
> - **BaRP**, by contrast:
>   - learns **directly under bandit feedback**,
>   - supports **per-request trade-off control** via the preference vector,
>   - and avoids dependence on the full performance matrix.
>
> We will revise Sec. 5 to more clearly acknowledge the adaptability of prior methods while clarifying these practical distinctions.
>
> ---
>
> ## **3. OOD Evaluation — Cost/Accuracy Trade-off (Requested Numbers)**
>
> The reviewer asked whether BaRP’s strong OOD performance is simply due to selecting the largest LLM more often. To address this, we provide the explicit cost/accuracy values for two OOD tasks (preference: *w_q = w_c = 0.5*).
>
> ### **OOD Cost–Accuracy Results**
>
> | **Task** | **BaRP Score ↑** | **BaRP Cost ↓** | **Largest Score ↑** | **Largest Cost ↓** |
> |---------|------------------|------------------|-------------------|------------------|
> | **Hellaswag** | **0.8437** | **0.00127** | 0.8396 | 0.00216 |
> | **MBPP** | **0.7176** | **0.00577** | 0.6862 | 0.00938 |
>
> ### **Key Takeaways**
> - BaRP achieves **higher accuracy** *and* **~40–50% lower cost** on both tasks.
> - This indicates BaRP is **not** simply defaulting to the largest model.
> - Instead, BaRP learns **task-dependent routing patterns**, often preferring mid-sized models that perform well on specific prompt types.
> - We will include these results in the revised paper.
>
> ---
>
> We sincerely appreciate the reviewer’s helpful comments, which will directly strengthen the paper, and we will incorporate all of these enhancements in the final revision.

---

### Meta-Review · Area_Chair_YxCT · 2026-01-04

**Summary:**

This paper formulates LLM routing as a contextual bandit problem and proposed a bandit-based algorithm that supports preference-tunable inference.

The reviewers raised the following key concerns. First, the empirical evaluation was considered insufficient, particularly due to limited results on recent benchmarks and missing comparison against other baselines (e.g., existing predictive routers that can be easily adapted online). Second, reviewers noted the absence of online evaluation results, which is especially relevant given the bandit formulation. Third, the submission didn't include theoretical results or insights to support the proposed method. Finally, there were concerns regarding clarity and paper presentation.

**Reviewer Concerns:**

The rebuttal addressed concerns on paper presentations and clarified several points raised by the reviewers. It also partially addressed concerns regarding insufficient evaluation by providing additional results on the SPROUT benchmark; however, the rebuttal didn't include clear comparisons against other baselines.

The rebuttal didn't address concerns regarding missing online evaluation results or the lack of theoretical results/intuitions. While the authors stated that online learning dynamics would be provided, no such results were included during the rebuttal.

**Reviewer Scores:**

If the reviewers had been able to participate fully in the discussion, I believe Reviewer aVZK would likely have increased their score from 2 to 4. The other reviewers' scores would likely have remained unchanged at 4.

---

### Decision · Program_Chairs · 2026-01-26

Reject